# Epidemiological and clinical characteristics of scrub typhus in Guizhou Province, China: An outbreak study of scrub typhus

Jia He[1], Qing Ma[2], Zhongqiu Teng[1], Jingzhu Zhou[2], Na Zhao[1], Wenqin Liang[2], Miao Lu[1], Shijun Li[2], Tian Qin[1]*

1 National Key Laboratory of Intelligent Tracking and Forecasting for Infectious Diseases, National Institute for Communicable Disease Control and Prevention, Chinese Center for Disease Control and Prevention, Beijing, People's Republic of China, 2 Guizhou Center for Disease Control and Prevention, Guizhou, People's Republic of China

These authors contributed equally to this work.
* qintian@icdc.cn

**Data Availability Statement:** The authors confirm that all data underlying the findings are fully available without restriction. All relevant data are within the paper and its Supporting Information

## Abstract

The reported cases of scrub typhus (ST) have continued to escalate, with outbreaks occurring regionally in China. These pose an increasing public health threat at a time when public health has been overwhelmed. During the period from July to August 2022, in Rongjiang County, Guizhou Province, China, 13 out of 21 fever patients were diagnosed with scrub typhus, based on epidemiological investigation and blood test analysis. The major clinical symptoms of these patients showed fever, chills, headache, eschar, fatigue and pneumonia, which were accompanied by a rise in C-reactive protein, neutrophils, alanine transaminase (ALT) and aspartate aminotransferase (AST). Furthermore, nearly half of them exhibited abnormal electrocardiogram activity. Through semi-nested PCR, Sanger sequencing and phylogenetic tree construction, the Karp strain of *Orientia tsutsugamushi* (*O. tsutsugamushi*) was confirmed as the pathogen causing ST in Rongjiang County, which shared the same evolutionary branch with *O. tsutsugamushi* isolated from wild mouse liver or spleen, indicating that the wild mouse plays an important role in transmitting the disease. In contrast to the sporadic cases in the past, our study is the first to disclose an epidemic and the corresponding clinical characteristics of ST in Guizhou province, which is of great significance for the prevention and treatment of regional illnesses.

## Author summary

In this study, 13 cases of ST were identified through clinical and laboratory diagnosis in Rongjiang County, which is the first outbreak of the disease in Guizhou province, China. It is worth noting that ST was caused by the Karp strain, which can be transmitted by wild mice, and the infected population generally had a clear history of outdoor activity. In addition to the typical clinical symptoms such as fever, chills and pneumonia, nearly half of patients with ST also exhibited abnormal electrocardiogram activity. In conclusion, our study presents a method to confirm ST through the combination of clinical and laboratory

files.

**Funding:** This work was supported by the Science Foundation for the State Key Laboratory for Infectious Disease Prevention and Control from China under Grant number 2022SKLID209 (awarded to TQ) and 2019SKLID403 (awarded to TQ), the Public Health Service Capability Improvement Project of the National Health Commission of the People's Republic of China under Grant number 2100409002 (awarded to TQ) and the National Key Research and Development Program of China under Grant number 2021YFC2301202 (awarded to TQ). The funders had no role in study design, data collection and analysis, decision to publish, or preparation of the manuscript.

**Competing interests:** The authors have declared that no competing interests exist.

diagnosis, which provides a reference for the development of effective strategies and measures to tackle ST, and has important public health significance.

## Introduction

Scrub typhus (henceforth referred to here as ST) is an acute natural epidemic disease caused by *Orientia tsutsugamushi*. Wild rodents, especially various species of mice, are the source of transmission for the disease. Chigger mites are the vectors of the disease, and humans are typically infected through the bites of chigger mites carrying the pathogen [1].

Scrub typhus has emerged as a serious public health problem in the Asia-Pacific region, affecting several countries such as China, Korea, Japan, India, Indonesia, Thailand, Sri Lanka and the Philippines [2]. ST is a potentially fatal infection that is estimated to threaten one billion people in the Asia-Pacific region [3]. The initial symptoms manifested in an infected patient are fever, headache, chills and cough. The disease may progress to pneumonia and, in the most severe cases, result in death due to multiple organ failure and other complications [4]. Scrub typhus is difficult to distinguish from other febrile illnesses because of the similarity in patients presenting with clinical signs and symptoms to other diseases. Without appropriate treatment, the mortality rate of patients with ST may be as high as 30–70%, while the median mortality rate of untreated patients is 6%, compared to 1.4% for those who receive treatment [5]. In China, scrub typhus is mainly endemic in the north, south and southwest of the country [6], which includes the Guangdong, Yunnan, Anhui, Guangxi, Fujian, Jiangsu, Shandong and Jiangxi provinces [7].

This study details 13 cases of confirmed scrub typhus, including 1 fatal case, and is the first reported outbreak of scrub typhus in the Guizhou Province. We verified that the outbreak of scrub typhus was caused by the Karp strain, which is of great significance to public health. Furthermore, we provide a reference for the development of effective strategies and measures in tackling scrub typhus.

## Materials and methods

### Ethics statement

The study was approved by the ethics committee of the Chinese Center for Disease Control and Prevention and National Institute for Communicable Disease Control and Prevention (ICDC-202115). Written consent was obtained from all patients prior to the study.

### Sample collection

From July to August 2022, 21 cases of fever in residents were identified at Rongjiang County People's Hospital of Guizhou Province (Fig 1). Further, epidemiological investigation of the cases was conducted through a questionnaire aimed at capturing the clinical characteristics of the patients. Blood samples or double blood samples were collected from 7 patients, and 1 bronchoalveolar lavage fluid (BALF) specimen was taken from the deceased patient. In the same period, a total of 84 wild mouse samples (liver or spleen) were collected by the Guizhou Center for Disease Control and Prevention from various locations in Rongjiang County, Guizhou Province. Fig 2 shows the number of cases and mice collected in each period. A red dot in the figure represents the collection period for a positive mouse.

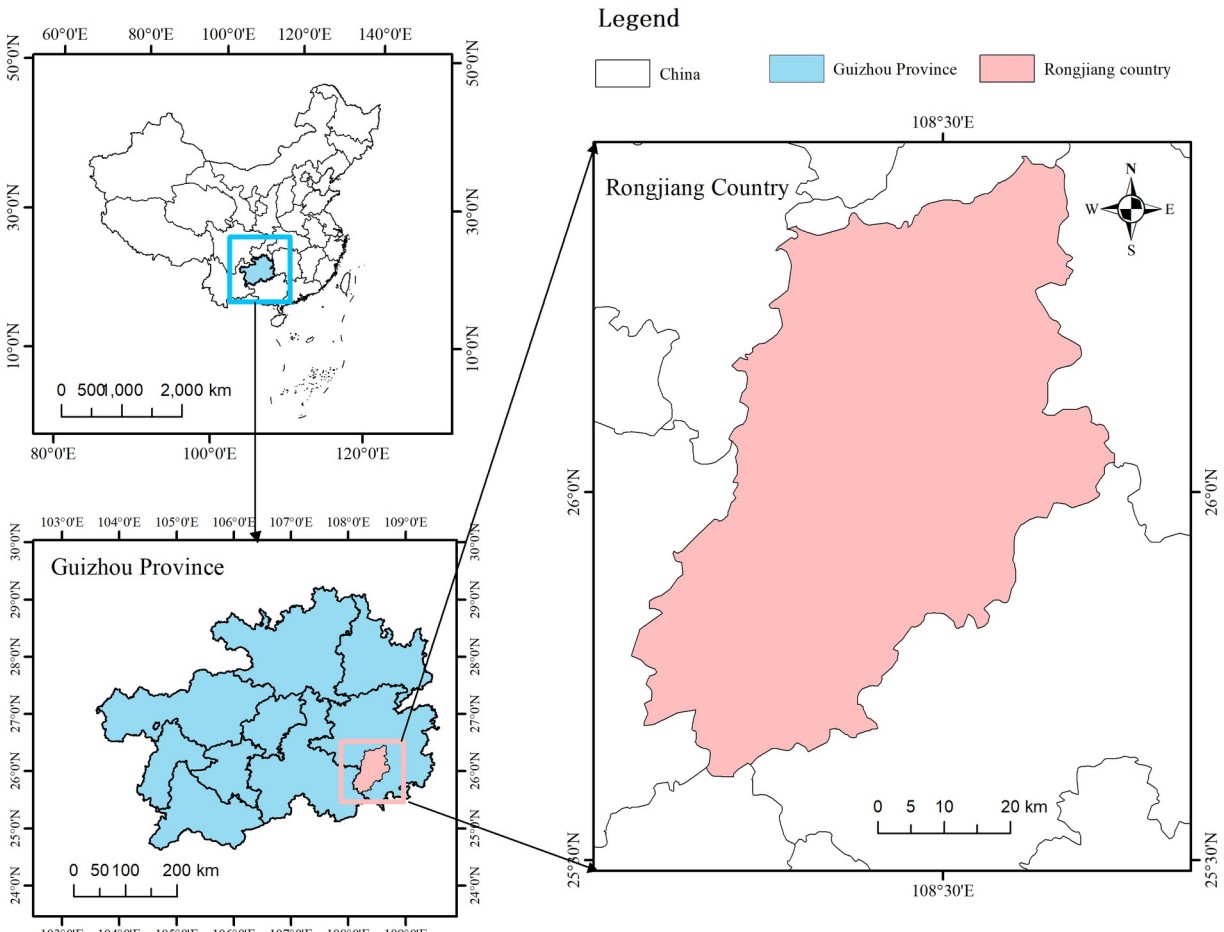

**Fig 1. Map showing the location of patients with scrub typhus infection in Rongjiang County, Guizhou Province, China.** The map was constructed using ArcGIS 10.8 software. The basemap shapefiles were downloaded from national platform for common geospatial information services (tianditu.gov.cn).

## Case definition

The classification for suspected, clinically-confirmed and laboratory-confirmed cases was based on the technical guidelines for the prevention and control of scrub typhus issued by the Chinese CDC [8]. A summary of the case definitions is available in S1 Text.

## PCR and Phylogenetic analysis

EDTA blood samples were directly subjected to DNA extraction using a QIAamp DNA Blood Mini Kit following the manufacturer's instructions. The gene encoding the 56-kDa protein (56-kDa gene), the 47-kDa protein (47-kDa gene) and 16S rRNA (16S rRNA gene) of *O. tsutsugamushi* was amplified from the DNA samples and PCR was used for diagnosis of scrub typhus [9,10]. PCR products with the target DNA bands were sent to Tianyi Huiyuan Biotechnology Company (Beijing, China) for sequencing. The obtained DNA sequences were compared to those previously published in GenBank (http://blast.ncbi.nlm.nih.gov/) using BLAST. The sequences reported in this paper have been deposited in GenBank, the accession numbers are provided in S1 Table (OR501559-OR501565, OR513496-OR513508).

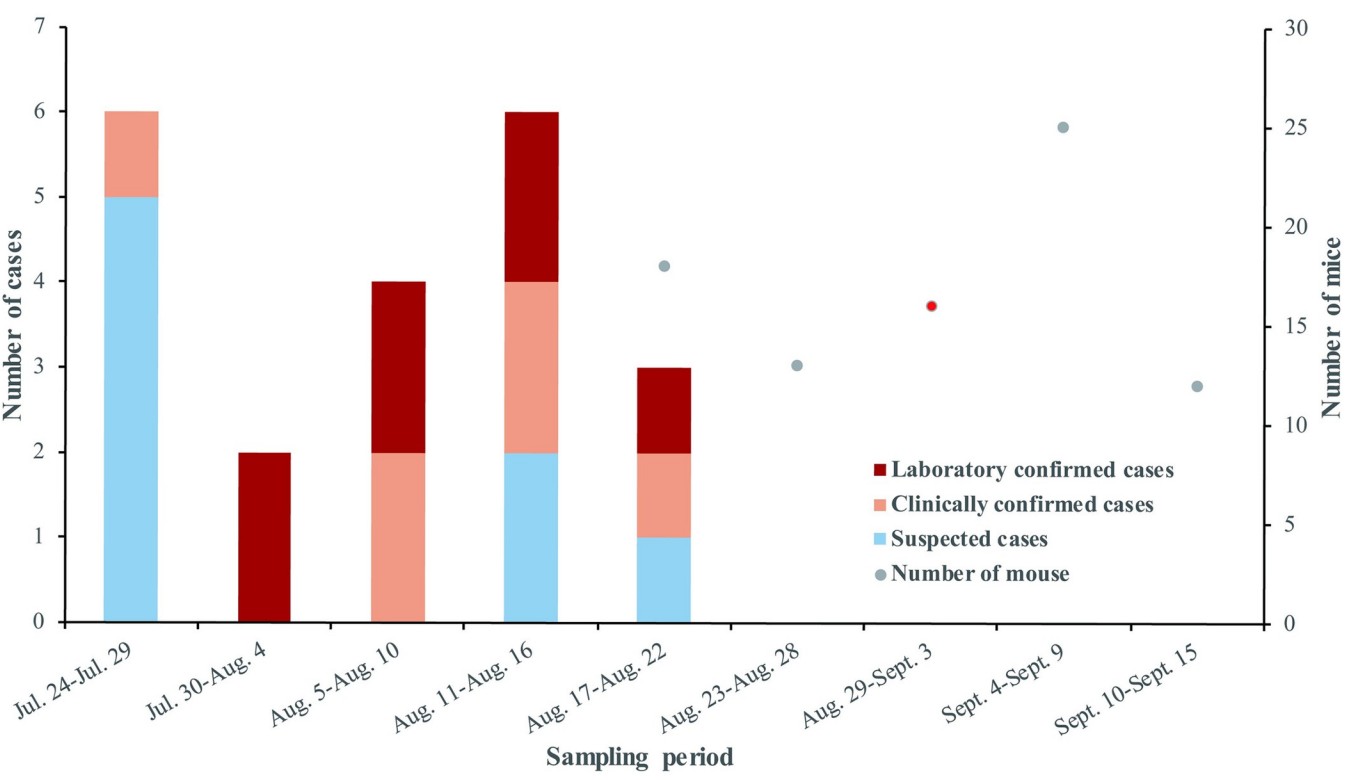

**Fig 2. The number of cases and mice collected in each period.**

The sequences were aligned and analyzed using the multiple sequence alignment program ClustalW (default parameters) as implemented in MEGA, version 11.0. Phylogenetic trees were generated based on the aforementioned three gene sequences, using the neighbor-joining method with 1000 replicates for bootstrap testing.

### Serologic testing

Serum-specific IgG or IgM antibodies against *O. tsutsugamushi* were detected in patient sera using the commercial immunofluorescence assay (IFA) kits (Fuller laboratories, CA, USA). In this study, the end point titers for IgM and IgG sera were detected using the serial two-fold dilution method. A single serum IgM antibody titer of not less than 1:32 or IgG antibody titer of not less than 1:64 was judged to be a positive infection. A 4-fold or greater increase in IgG titer between acute and convalescent sera was also judged as a positive infection.

## Results

### PCR Assay and Sequencing

The 7 whole blood samples and 1 BALF sample were collected from 21 patients with fever and a history of outdoor activity. Based on the results from the semi-nested PCR, 6 samples from the patients 1, 3, and 5–8 tested positive for infection with *O. tsutsugamushi*. The 56-kDa, 47-kDa and16S rRNA gene fragments were amplified from the 5 samples from patients 1, 3, 5, 7 and 8. Only the 16S rRNA and 56-kDa gene fragment were amplified from the sample of patient 6, while the 47-kDa gene fragment was not detected. No gene fragments were amplified from the samples of patients 2 and 4 (Table 1).

**Table 1. Serological and microbiological diagnostic data of 7 patients infected with scrub typhus.**

| Patient No. | Sample type | PCR amplification | | | IFA IgM | IFA IgG | |
|---|---|---|---|---|---|---|---|
| | | 56kDa | 16S rRNA | 47kDa | AP | AP | CP |
| 1 | Blood | + | + | + | 1:80 | 1:128 | 1:2048 |
| 2 | Blood | - | - | - | - | - | - |
| 3 | Blood | + | + | + | 1:160 | 1:256 | 1:2048 |
| 4 | Blood | - | - | - | 1:160 | 1:256 | 1:2048 |
| 5 | Blood | + | + | + | 1:160 | 1:256 | 1:2048 |
| 6 | Blood | + | + | - | 1:80 | - | - |
| 7 | Blood | + | + | + | 1:160 | 1:256 | 1:2048 |
| 8 | BALF | + | + | + | NA | NA | NA |

AP, acute phase; CP, convalescent phase; NA, not available.

A total of 84 DNA samples were extracted from the liver or spleen of wild mice and detected using the PCR specific for *O. tsutsugamushi*. Only one sample tested positive (1/84, 1.2%). The 56-kDa, 47-kDa, and 16S rRNA gene fragments of *O. tsutsugamushi* were amplified from the mouse sample using PCR. The obtained gene fragments were sequenced, spliced, and aligned. Phylogenetic trees were generated based on the three gene sequences using the neighbor-joining method with 1000 replicates for bootstrap testing. As shown in Fig 3, phylogenetic analysis of the 56-kDa gene demonstrated that the strains detected in the present study were closely related to the Karp serotype strains. Our results also showed that the phylogenetic trees constructed using the 56-kDa, 47-kDa or 16S rRNA gene sequences were displayed similar topology. Simultaneously, *O. tsutsugamushi* strains from the patients and the single mouse sample were clustered together.

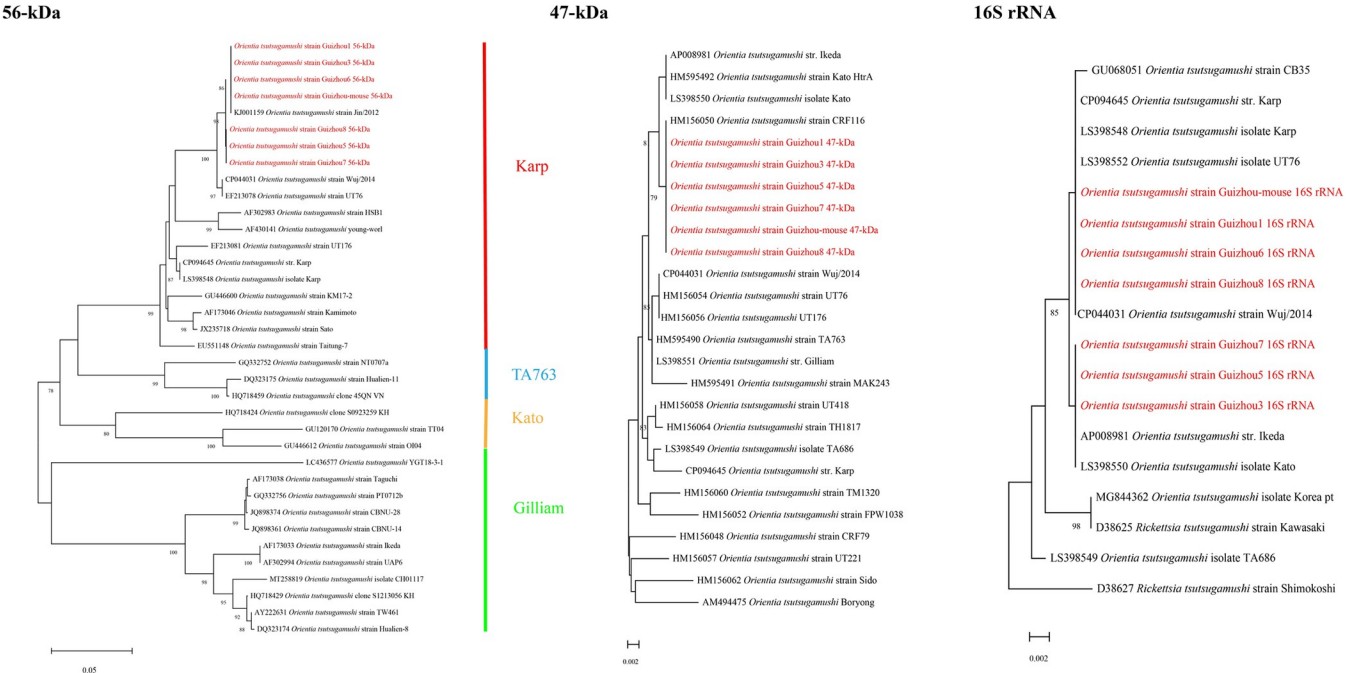

**Fig 3. Phylogenetic trees based on the nucleotide sequences of 56-kDa, 47-kDa, and 16S rRNA genes from *O. tsutsugamushi* strains.**

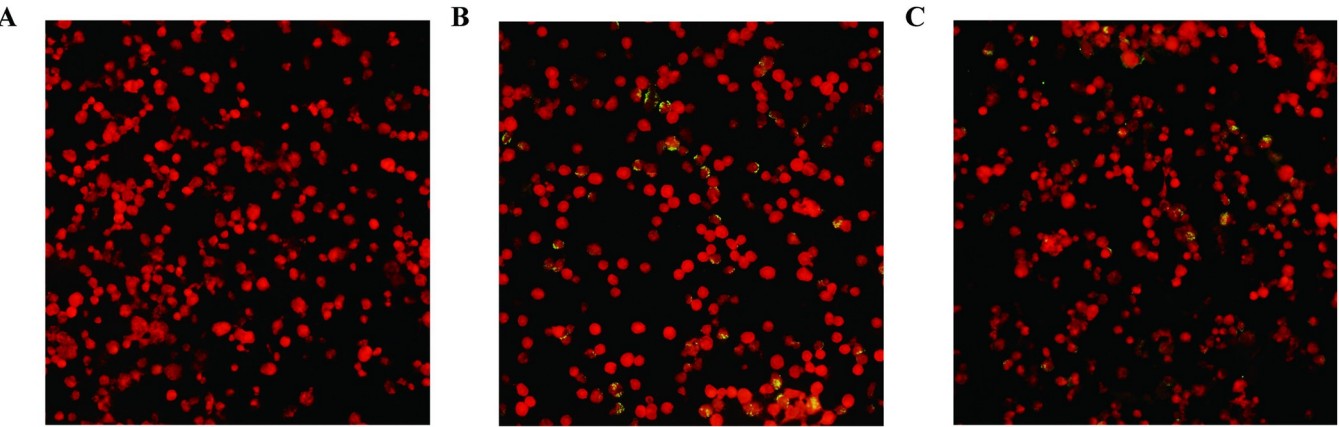

**Fig 4. Acute and convalescent serum samples were detected by indirect immunofluorescence assay (Samples from Patient 3).** (A) Negative control (B) Acute phase titer: 1:256. (C) Convalescent phase titer: 1:2048.

In the homology analysis of the 56-kDa gene, the homology between the *O. tsutsugamushi* strains detected in the present study and the *O. tsutsugamushi* Jin/2012 strain, Wuj2014 strain, and UT76 strain was 99.67%-100%, 98.34%-98.67% and 98.34%-98.67%, respectively. In the homology analysis of the 47-kDa gene, the homology between *O. tsutsugamushi* strains in the present study and *O. tsutsugamushi* CRF116, Wuj2014, and UT76 strain was 100%, 99.50%, and 99.50%, respectively. In the homology analysis of the 16S rRNA gene, the homology between *O. tsutsugamushi* strains in the present study and the *O. tsutsugamushi* Wuj2014 strain and UT76 strain was 99.88%-100% and 99.88%-100%, respectively. Furthermore, both the *O. tsutsugamushi* Jin/2012 strain and Wuj2014 strain have been found in Zhejiang Province.

## Serological tests

Both acute and convalescent phase serum samples from 7 patients were tested for IgM and IgG antibody titers using an indirect immunofluorescence assay. As shown in Table 1, 6 of the 7 patients had IgM antibody titers>1:32 and 5 patients had IgG antibody titers>1:64. The double serum samples from 5 patients exhibited a 4-fold increase in titer. The positive IFA results from patient 3 are shown in Fig 4. According to the *O. tsutsugamushi* specific PCR and/or IFA assays, seven patients were confirmed as having *O. tsutsugamushi* infection.

## Patients and clinical presentation

A total of 13 out of 21 febrile patients were confirmed as having scrub typhus, including the 6 cases confirmed by clinical diagnosis and 7 cases confirmed by laboratory diagnosis. All 13 patients had clear histories of outdoor activities. Out of the total confirmed cases, 11 (84.6%) were male and 12 (92.3%) were farmers. Their ages ranged from 24 to 71 years, with 8 (61.6%) aged between 30 and 49. The 12 (92.3%) of the 21 patients fully recovered but one (7.7%) died (Table 2).

In the 13 confirmed cases of scrub typhus, the main clinical symptoms exhibited were fever (100%), chills (84.6%), headache (76.9%), eschar (69.2%), asthenia (61.5%) and pneumonia (53.8%), followed by dizziness (46.2%), lymph node enlargement (46.2%), anorexia (38.5%), myalgia (38.5%) and tonsillitis (30.8%). Less than 30% patients had shortness of breath, vomiting, cough, diarrhea, and sore throat. Furthermore, laboratory findings indicated that more than half of the patients had increased C-reactive protein (92.3%), increased neutrophil count

**Table 2. Demographic data of the 13 patients with scrub typhus diagnosed by molecular tools in RongJiang Country, 2022.**

|  | Number of cases | Percentage (%) |
|---|---|---|
| **Field activity** |  |  |
| Fishing | 8 | 8/13(61.5%) |
| Field work | 4 | 4/13(30.8%) |
| Swimming | 1 | 1/13(7.7%) |
| **Gender** |  |  |
| Male | 11 | 11/13(84.6%) |
| Female | 2 | 2/13(15.4%) |
| **Occupation** |  |  |
| Farmer | 12 | 12/13(92.3%) |
| Teacher | 1 | 1/13(7.7%) |
| **Age (Year)** |  |  |
| <20 | 0 | 0 |
| 20–29 | 2 | 2/13(15.4%) |
| 30–39 | 4 | 4/13(30.8%) |
| 40–49 | 4 | 4/13(30.8%) |
| 50–59 | 2 | 2/13(15.4%) |
| >60 | 1 | 1/13(7.7%) |
| **Outcomes** |  |  |
| Full recovery | 12 | 12/13(92.3%) |
| Death | 1 | 1/13(7.7%) |

(92.3%), elevated ALT or AST (84.6%), electrolyte disorders (76.9%), lymphopenia (69.2%), eosinophilia (69.2%), and thrombocytopenia (61.5%). Abnormal electrocardiogram accounted for 46.2% (Table 3).

## Discussion

In recent years, the number of ST cases reported has increased annually and is no longer confined to the 'tsutsugamushi triangle' [11,12]. Scrub typhus outbreaks have been reported in many provinces and cities in China, including Liaoning, Shandong, Fujian, Zhejiang, Jiangsu, Guangdong and Tianjin [6]. Notably, ST has also become the most common vector-borne disease in southern China. However, prior to this outbreak, there have only been sporadic cases of scrub typhus in Guizhou province. This study is the first report of an outbreak of scrub typhus in Guizhou province, China.

One of the patients in this study who was diagnosed with scrub typhus was negativity by ST-specific PCR but exhibited IFA positivity. Previous reports have also demonstrated that negative PCR results do not exclude active infection with *O. tsutsugamushi*, as the presence of the pathogen in the blood may be temporary [13]. The IFA is considered the gold standard method for ST diagnosis but molecular diagnosis is particularly helpful in the acute infection stage where low antibody titers may yield equivocal or negative results [14]. Therefore, in the present study, the combination of semi-nested PCR and IFA was utilized to maximize the rate of diagnosis and confirm that ST cases were caused by the Karp strain, a strain that has also been detected in patients from the Yunnan and Hainan provinces [15,16]. The Kawasaki, Gilliam and Kato strains have also been reported in other regions of China [17–20].

The diagnosis of scrub typhus is primarily based on clinical manifestations and examination. Although the presence of eschar/ulcer is an important diagnostic clue for ST, it was

**Table 3. Clinical signs, symptoms, and laboratory findings of patients diagnosed with scrub typhus in RongJiang Country, 2022.**

|  | Number of cases | N, Percentage (%) |
|---|---|---|
| **Clinical Characterization** |  |  |
| Fever | 13 | 13/13(100%) |
| Chill | 11 | 11/13(84.6%) |
| Headache | 10 | 10/13(76.9%) |
| Eschar or ulcer | 9 | 9/13(69.2%) |
| Asthenia | 8 | 8/13(61.5%) |
| Pneumonia | 7 | 7/13(53.8%) |
| Dizziness | 6 | 6/13(46.2%) |
| Lymphadenopathy | 6 | 6/13(46.2%) |
| Anorexia | 5 | 5/13(38.5%) |
| Myalgia | 5 | 5/13(38.5%) |
| Tonsillitis | 4 | 4/13(30.8%) |
| Shortness of breath | 3 | 3/13(23.1%) |
| Vomiting | 3 | 3/13(23.1%) |
| Cough | 2 | 2/13(15.4%) |
| Diarrhea | 2 | 2/13(15.4%) |
| Sore throat | 2 | 2/13(15.4%) |
| **Laboratory findings** |  |  |
| CRP increased | 12 | 12/13(92.3%) |
| Neutrophilic leukocytosis | 12 | 12/13(92.3%) |
| Elevated ALT or AST | 11 | 11/13(84.6%) |
| Electrolyte disturbance | 10 | 10/13(76.9%) |
| Lymphocytopenia | 9 | 9/13(69.2%) |
| Eosinophil decrease | 9 | 9/13(69.2%) |
| Thrombocytopenia | 8 | 8/13(61.5%) |
| Abnormal electrocardiogram | 6 | 6/13(46.2%) |

absent in 30.8% (4/13) of the patients in this study. Thus, if there is reliance on this alone, it will lead to both under-diagnosis and misdiagnosis [21]. In this study, the patients were mainly male, middle-aged, and engaged in farming occupation, similar to the findings reported in previous studies [22–24]. However, Narang et al. demonstrated that females were at greater risk for the infection than males [25]. The clinical symptoms of scrub typhus are mainly characterized by fever, chills, headache, asthenia, dizziness, pneumonia, and lymphadenopathy, which are non-specific and could be attributed to other conditions. But in this study, there was a rare case of death due to scrub typhus pneumonia.

In addition, in the laboratory profile of scrub typhus patients, thrombocytopenia, liver function test (indicating damage) and electrolyte disorders are commonly used to evaluate the severity of patients but the initial electrocardiogram (ECG) manifestations are often overlooked. It is worth noting that 46.2% (6/13) of patients had an abnormal ECG in this study. Choi et al. showed that the severity of ST was greater in the abnormal ECG group compared to the normal ECG group [26]. Interestingly, our study and Sun et al. both found a low level of eosinophils, and further research is needed to explore the relationship between ST and eosinophils [27].

There are several possibilities as to how this outbreak of scrub typhus emerged. Firstly, the temperature and humidity during the outbreak period in this study were the climate risk windows of scrub typhus [28]. Secondly, the movement of migrant workers is also one of the

possible causes of increased likelihood of scrub typhus. Thirdly, since the monitoring capacity of medical institutions and clinicians for ST has continuously improved, which reduces the possibility of misdiagnosis and missed diagnoses.

Given that this is the first reported outbreak of scrub typhus in Guizhou Province, it is of great significance that there is continued efforts in basic research on the classification of *O. tsutsugamushi* strains prevalent in the region, the development of vaccines where feasible, and accurate diagnosis of scrub typhus using optimal modalities. In endemic areas of scrub typhus, health education programs should be strengthened to improve the awareness of the sources of infection, transmission routes, and symptoms of scrub typhus, with coordinated efforts to improve prevention and control methods. Simultaneously, regards future work, it is necessary to further control the rodents and vectors associated with scrub typhus, while strengthening the epidemiological surveillance and increased training of medical personnel to reduce the risk and complications of human infection.

## Supporting information

**S1 Data. The underlying numerical data for Fig 2.**
(XLSX)

**S2 Data. The underlying numerical data for Table 2.**
(XLSX)

**S3 Data. The underlying numerical data for Table 3.**
(XLSX)

**S1 Table. GenBank numbers of Orientia tsutsugamushi sequences obtained in this study.**
(PDF)

**S2 Table. Primers used for the Scrub typhus.**
(PDF)

**S1 Text. Case definition.**
(PDF)

## Acknowledgments

We thank all the participants in the study.

## Author Contributions

**Conceptualization:** Jia He, Qing Ma, Tian Qin.

**Data curation:** Jia He, Qing Ma, Na Zhao.

**Funding acquisition:** Tian Qin.

**Investigation:** Jingzhu Zhou, Wenqin Liang, Shijun Li.

**Methodology:** Qing Ma, Zhongqiu Teng, Miao Lu.

**Software:** Jia He, Zhongqiu Teng.

**Supervision:** Tian Qin.

**Writing – original draft:** Jia He, Qing Ma.

**Writing – review & editing:** Zhongqiu Teng, Na Zhao, Tian Qin.

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
