## [Decision Letter · Decision Letter 0]

27 Dec 2023

Dear Ms. He,

Thank you very much for submitting your manuscript "Epidemiological and clinical characteristics of scrub typhus in Guizhou Province, China: An outbreak study of scrub typhus" for consideration at PLOS Neglected Tropical Diseases. As with all papers reviewed by the journal, your manuscript was reviewed by members of the editorial board and by several independent reviewers. The reviewers appreciated the attention to an important topic. Based on the reviews, we are likely to accept this manuscript for publication, providing that you modify the manuscript according to the review recommendations. 

Two reviewers have gone through the paper and recommend some changes. There is a recommendation of revising the length of the paper to a shorted format by one of the reviewers. I suggest that you go through the comments and submit a revised manuscript including only the relevant aspects of this new focus of scrub typhus in this area.

Sincerely,

Manisha Biswal

Academic Editor

Nigel Beebe

Section Editor

Two reviewers have gone through the paper and recommend some changes. There is a recommendation of revising the length of the paper to a shorted format by one of the reviewers. I suggest that you go through the comments and submit a revised manuscript including only the relevant aspects of this new focus of scrub typhus in this area.

Reviewer's Responses to Questions

**Key Review Criteria Required for Acceptance?**

**Methods**

-Are the objectives of the study clearly articulated with a clear testable hypothesis stated?

-Is the study design appropriate to address the stated objectives?

-Is the population clearly described and appropriate for the hypothesis being tested?

-Is the sample size sufficient to ensure adequate power to address the hypothesis being tested?

-Were correct statistical analysis used to support conclusions?

-Are there concerns about ethical or regulatory requirements being met?

Reviewer #1: As an outbreak report, a lot of the above are not relevant to this manuscript. For those relevant, such as statistical analysis and ethical clearances, the methods are acceptable.

Reviewer #2: -Are the objectives of the study clearly articulated with a clear testable hypothesis stated? - Yes

-Is the study design appropriate to address the stated objectives? - Yes. Need more information. Comments provided.

-Is the population clearly described and appropriate for the hypothesis being tested? - Yes. Need more information. Comments provided.

-Is the sample size sufficient to ensure adequate power to address the hypothesis being tested? - NA

-Were correct statistical analysis used to support conclusions? - NA

-Are there concerns about ethical or regulatory requirements being met? - Yes

**Results**

-Does the analysis presented match the analysis plan?

-Are the results clearly and completely presented?

-Are the figures (Tables, Images) of sufficient quality for clarity?

Reviewer #1: Yes, results presented appropriately.

Reviewer #2: -Does the analysis presented match the analysis plan? - Yes

-Are the results clearly and completely presented? - Yes. 

-Are the figures (Tables, Images) of sufficient quality for clarity? - Need better quality images

**Conclusions**

-Are the conclusions supported by the data presented?

-Are the limitations of analysis clearly described?

-Do the authors discuss how these data can be helpful to advance our understanding of the topic under study?

-Is public health relevance addressed?

Reviewer #1: Conclusions drawn and presented adequately.

Reviewer #2: -Are the conclusions supported by the data presented? - Need more information. Comments provided.

-Are the limitations of analysis clearly described? - No

-Do the authors discuss how these data can be helpful to advance our understanding of the topic under study? - Yes

-Is public health relevance addressed? - Yes

**Editorial and Data Presentation Modifications?**

Reviewer #1: Please attend to the following: 

1. Correct all typo errors after a thorough read through again by all authors 

2. Use of words and digits to be made uniform throughout (like 1 or one)

3. Improve the overall presentation

Reviewer #2: (No Response)

**Summary and General Comments**

Reviewer #1: (No Response)

Reviewer #2: The study focused on important concerns related to febrile illness in the study region. Relevant tools to investigate the cases and animal hosts were performed to link the association but the vector association was not investigated. Suspected and clinical cases along with lab-confirmed cases need to be defined with better clarity.

PLOS authors have the option to publish the peer review history of their article (what does this mean?). If published, this will include your full peer review and any attached files.

Reviewer #1: Yes: Tshokey

Reviewer #2: Yes: Sivanantham Krishnamoorthi

Figure Files:

Data Requirements:

Reproducibility:

References

---

## [Editor Report · Decision Letter 1]

2 Feb 2024

Dear Ms. He,

We are pleased to inform you that your manuscript 'Epidemiological and clinical characteristics of scrub typhus in Guizhou Province, China: An outbreak study of scrub typhus' has been provisionally accepted for publication in PLOS Neglected Tropical Diseases.

Best regards,

Manisha Biswal

Academic Editor

Nigel Beebe

Section Editor

The authors have adequately addressed the queries of the reviewers.

---

## [Editor Report · Acceptance letter]

20 Feb 2024

Dear Ms. He,

We are delighted to inform you that your manuscript, "Epidemiological and clinical characteristics of scrub typhus in Guizhou Province, China: An outbreak study of scrub typhus," has been formally accepted for publication in PLOS Neglected Tropical Diseases.

Best regards,

Shaden Kamhawi

co-Editor-in-Chief

Paul Brindley

co-Editor-in-Chief
